# Discovery of Benzopyrone-Based Candidates as Potential Antimicrobial and Photochemotherapeutic Agents through Inhibition of DNA Gyrase Enzyme B: Design, Synthesis, In Vitro and In Silico Evaluation

**DOI:** 10.3390/ph17091197

**Published:** 2024-09-11

**Authors:** Akram Abd El-Haleem, Usama Ammar, Domiziana Masci, Sohair El-Ansary, Doaa Abdel Rahman, Fatma Abou-Elazm, Nehad El-Dydamony

**Affiliations:** 1Pharmaceutical Chemistry Department, College of Pharmaceutical Sciences and Drug Manufacturing, Misr University for Science and Technology, Al-Motamayez District, 6th of October City P.O. Box 77, Egypt; sohir.alansary@must.edu.eg (S.E.-A.); nehad.eldydamony@must.edu.eg (N.E.-D.); 2School of Applied Sciences, Edinburgh Napier University, Sighthill Campus, 9 Sighthill Court, Edinburgh EH11 4BN, UK; 3Department of Basic Biotechnological Sciences, Intensivological and Perioperative Clinics, Catholic University of the Sacred Heart, Largo Francesco Vito 1, 00168 Rome, Italy; domiziana.masci@unicatt.it; 4Department of Pharmaceutical Chemistry, Faculty of Pharmacy, Cairo University, Kasr El-Aini Street, Cairo 11562, Egypt; 5Department of Microbiology and Immunology, College of Pharmaceutical Sciences and Drug Manufacturing, Misr University for Science and Technology, Al-Motamayez District, 6th of October City P.O. Box 77, Egypt; fatma.alsayed@must.edu.eg

**Keywords:** furobenzopyrones, antimicrobial activity, photosensitizing activity, molecular docking, in silico prediction

## Abstract

Bacterial DNA gyrase is considered one of the validated targets for antibacterial drug discovery. Benzopyrones have been reported as promising derivatives that inhibit bacterial DNA gyrase B through competitive binding into the ATP binding site of the B subunit. In this study, we designed and synthesized twenty-two benzopyrone-based derivatives with different chemical features to assess their antimicrobial and photosensitizing activities. The antimicrobial activity was evaluated against *B. subtilis*, *S. aureus*, *E. coli*, and *C. albicans*. Compounds **6a** and **6b** (rigid tetracyclic-based derivatives), **7a**-**7f** (flexible-linker containing benzopyrones), and **8a**-**8f** (rigid tricyclic-based compounds) exhibited promising results against *B. subtilis*, *S. aureus*, and *E. coli* strains. Additionally, these compounds demonstrated photosensitizing activities against the *B. subtilis* strain. Both in silico molecular docking and in vitro DNA gyrase supercoiling inhibitory assays were performed to study their potential mechanisms of action. Compounds **8a**-**8f** exhibited the most favorable binding interactions, engaging with key regions within the ATP binding site of the DNA gyrase B domain. Moreover, compound **8d** displayed the most potent IC_50_ value (0.76 μM) compared to reference compounds (novobiocin = 0.41 μM and ciprofloxacin = 2.72 μM). These results establish a foundation for structure-based optimization targeting DNA gyrase inhibition with antibacterial activity.

## 1. Introduction

The dramatic upsurge in the prevalence of microorganisms, including bacteria, viruses, and fungi, is considered a potential warning of significant health risks, potentially leading to widespread infections and pandemics. These microorganisms can infect people through various means such as water, food, airborne transmission, contact with a host, and exposure to contaminated surfaces. Therefore, preventing the rapid spread of these harmful microorganisms is crucial in order to prevent the onset of infectious diseases [1,2]. Although approved antibiotics have been widely used, the rise of bacterial resistance to the majority of these commonly prescribed drugs has created a pressing demand for new and efficient antibacterial therapeutic options. Therefore, the primary obstacle in addressing this need is finding potent antibacterial substances that not only have a broad spectrum of effectiveness but also exhibit a better safety profile [3,4,5,6].

One of the key targets for antibacterial agents is the DNA gyrase enzyme. DNA gyrase belongs to a family of ubiquitous and essential ATP-dependent enzymes, known as type II topoisomerases, which are responsible for controlling the level of supercoiling of DNA in cells [7]. The importance of bacterial DNA gyrase during the replication process lies in its ability to maintain the topology of the DNA duplex [8]. DNA gyrase is responsible for introducing negative supercoils into DNA through ATP hydrolysis [9,10,11]. DNA gyrase enzyme consists of two major subunits, GyrA and GyrB, which together form a functional heterodimer in GyrA2GyrB2 stoichiometry. The GyrA subunit primarily facilitates the breakage of the bacterial DNA, whereas the GyrB subunit exhibits ATPase activity (Figure 1). In the absence of ATP, DNA gyrase can only relax supercoiled DNA but cannot introduce negative supercoils [10,11]. Therefore, DNA gyrase represents one of the most clinically important biological targets for antibacterial agents due to its absence in the mammalian organism and its crucial role in the bacterial DNA replication cycle [8,12].

Different fused heterocyclic-based derivatives have been reported to be used as antibacterial agents through the inhibition of the bacterial DNA gyrase enzyme such as fluoroquinolones I, benzothiazoles II, and benzopyrones III, IV (Figure 2A) [1,15].

Benzopyrones and furobenzopyrones have extended therapeutic applications through the ability to exert non-covalent interactions with the active sites of different biological targets. Therefore, benzopyrones and furobenzopyrones exhibit a wide range of pharmacological activities such as anticoagulant [16], anti-neurodegenerative [17], antioxidant [18], anticancer [19,20], and antimicrobial activities [21,22,23].

Recent reports highlight GyrB inhibitors from diverse chemical classes, including indazole, pyrazole, benzimidazole, phenol, and indolinone [14,24,25,26,27]. The benzopyrones such as novobiocin III and clorobiocin (IV, aminobenzopyrones) are natural antibiotics derived from Streptomyces species and they exhibit their antibacterial effect through inhibition of DNA gyrase enzyme (GyrB subunit) [1]. They are considered competitive inhibitors, targeting the ATP-binding site on the GyrB subunit [28]. DNA gyrase B inhibitors are not currently in clinical use. Novobiocin III was withdrawn due to severe side effects and toxicity issues [29]. Therefore, additional medicinal chemistry-based research activities should be conducted to develop new drug candidates targeting the DNA gyrase B subunit through competitive inhibition of the ATP binding site.

Notably, benzopyrone-based scaffolds are incorporated into various derivatives as effective photochemotherapeutic agents (PCT), including psoralen V, carbethoxypsoralen VI, pyridopsoralens VII, and benzofurobenzopyrones VIII (Figure 2B) [30]. The combination of psoralen (V, a tricyclic linear furobenzopyrone) and UV-A (320–400 nm) has been discovered to be an effective photochemotherapy targeting DNA (PUVA therapy) [30,31]. Combination therapy involving photochemotherapeutic strategies has been reported to show a synergetic effect in antimicrobial activities [32,33].

Consequently, the goal of the current study was to conduct a series of molecular modifications and structural developments of furan-fused benzopyrone as a potential scaffold in order to afford first-in-class candidates with antimicrobial activity through a dual synergetic mechanism of action, inhibition of the DNA gyrase enzyme and photoreaction with DNA (Figure 3). Hence, in the current study, we describe the design and synthesis of a novel series of tricyclic (angular furobenzopyrones) and tetracyclic (tetrahydrobenzopyrone and tetrahydrobenzofurobenzopyrone) derivatives. Furthermore, all synthesized compounds were evaluated for their cellular antimicrobial and photochemotherapeutic activities. The mechanism of antimicrobial activity was also screened and evaluated through an in vitro DNA gyrase supercoiling inhibitory assay, to identify the IC_50_ values, and in silico molecular docking screening to explore the possible binding interactions with the key amino acid residues at the ATP binding site of the target biological domain. Additionally, the ADME profiles of promising candidates were predicted to assess the drug-likeness of the designed derivatives.

## 2. Results and Discussion

### 2.1. Chemistry

The synthesis of the proposed benzopyrones, including furobenzopyrones, tetrahydrobenzo-, and benzofurobenzopyrones was accomplished as shown in Figure 1 and Figure 2. Starting compounds, 6-ethyl-7-hydroxy-4-substituted-2*H*-1-benzopyran-2-ones **1a** and **1b** were synthesized according to the reported procedure by Christian Vaccarin et al., 2022 [34].

7-Hydroxybenzopyrones **1a** and **1b** were heated under reflux in the presence of anhydrous potassium carbonate with appropriate α-haloketone in dry acetone yielded ether derivatives **2a, 2b**, **4a** and **4b** (Figure 1). The negative ferric chloride test confirmed the absence of the phenolic hydroxyl group of compounds **1a** and **1b**. In addition, the desired structures were confirmed by spectral and analytical data. ^1^H NMR spectra of compounds **2a** and **2b** revealed the appearance of doublet, singlet, and quartet signals at 1.48–1.50, 2.21–2.24, and 5.16–5.21 ppm attributed to OCHCH_3_, COCH_3_, and OCHCH_3_ of the formed 3-oxobutan-2-yloxy moiety, respectively. ^1^H NMR spectra illustrated the appearance of multiplet and triplet signals between 1.54–5.27 ppm referred to cyclohexanone protons of **4b**.

Refluxing ether derivatives **2a**, **2b**, **4a**, or **4b** with sodium hydroxide in absolute ethanol was carried out followed by subsequent acidification resulted in their cyclization to the corresponding furo[2,3-*h*]benzopyran-5-ones **3a** and **3b** or tetrahydrobenzofurobenzopyrones **5a** and **5b**, respectively, (Figure 1). The structures of the synthesized compounds were characterized by spectral and analytical data. ^1^H NMR spectra showed the disappearance of singlet signals of the H-8 aromatic protons of benzopyrone at 6.84–7.02 ppm. Further, cyclization was assured by the presence of one singlet signal at 7.06–7.37 or 6.99–7.18 ppm corresponding to H-8 proton of furobenzopyrone in compounds **3a** and **3b**, or H-5 in tetrahydrofurobenzopyrone compounds **5a** and **5b**, respectively.

The use of DDQ in dry benzene for dehydrogenation of tetrahydrobenzo- compounds **5a** and **5b** yielded the corresponding benzofurobenzopyrones **6a** and **6b** (Figure 1). ^1^H NMR spectra assured the absence of two multiplet signals at 1.84–1.99 and 2.77–3.01 ppm attributed to tetrahydrobebzo- protons for compound **5a**. Alternatively, two triplet and two doublet signals for compound **6a** appeared at 7.42, 7.53, 7.62, and 8.39 ppm, respectively attributed to H-9, H-10, H-8, H-11 aromatic protons, respectively, thus confirming dehydrogenation.

Etherification of 7-hydroxybenzopyrones **1a** and **1b** with appropriate phenacyl bromide derivative and anhydrous potassium carbonate afforded compounds **7a**-**7f** (Figure 2). The negative ferric chloride test in addition to spectral and analytical data inferred the formation of the new ether derivatives. ^1^H NMR showed the presence of a singlet signal at 5.69–5.83 ppm attributed to OCH_2_CO protons together with the increased number of aromatic protons assured etherification.

Ether derivatives **7a**-**7f** were cyclized through refluxing with freshly prepared sodium ethoxide solution, followed by subsequent acidification thus affording angular furobenzopyrones **8a**-**8f** (Figure 2, Table 1). All spectral and analytical data ensured the structures of the synthesized compounds. ^1^H NMR spectra showed the disappearance of singlet peaks attributed to OCH_2_CO and H-8 aromatic proton of benzopyrone assured cyclization and formation of angular furobenzopyrone. Instead, the appearance of singlet peak at 8.22–8.34 ppm was assigned to H-2 aromatic proton.

### 2.2. Antimicrobial Assay

Antimicrobial screening assay was conducted for the synthesized compounds against four microorganisms (*B. subtilis*, *S. aureus*, *E. coli*, and *C. albicans*) using xanthotoxin, tobramycin, and amphotericin B as standards. The antimicrobial activities were evaluated by measuring the diameters of the inhibition zones (in mm) (Table 2). The results showed that most compounds exhibited higher activity than xanthotoxin against *B. subtilis*, while compounds **7c**, **7e** and **7f** were either equal in activity or inactive, respectively. The tetracyclic furobenzopyrone derivatives **6a** and **6b** might demonstrate significant antibacterial activity.

Most compounds showed good antibacterial activity against *S. aureus*, though compounds **7c**-**7f** were inactive. Almost all furobenzopyrones exhibited significant antibacterial activity against *S. aureus*, suggesting the crucial role of furan moiety, combined with benzopyrone, in antimicrobial activity. Additionally, the significant antimicrobial activity of rigid tetracyclic derivatives **6a** and **6b** against *S. aureus* might highlight the impact of their tetracyclic structure.

The antimicrobial results revealed that the aromatic tetracyclic-based derivatives **6a** and **6b** exhibited antibacterial activity against *E. coli*. In contrast, benzopyrone derivatives **7a**, **7b**, **7e** and **7f** showed reasonable antibacterial activity against *E. coli* compared to tobramycin, while compounds **2a**, **2b**, **4a**, and **4b** were inactive. This suggests the importance of extending the core scaffold with an additional phenyl ring. Interestingly, angular furobenzopyrone derivatives **8a**-**8e** showed significant antibacterial activity against *E. coli* higher than that of tobramycin. This might indicate the effect of decorating the furan ring with an additional phenyl ring. Conversely, the presence of *p*-methoxyphenyl substitution of furobenzopyrones resulted in compounds **8c** and **8f** having decreased activity against *E. coli* compared to unsubstituted phenyl or *p*-methylphenyl derivatives **8a**, **8b**, **8d**, and **8e**.

While the benzopyrone derivative **4a** showed excellent antifungal activity comparable to amphotericin B, most benzopyrone derivatives were only slightly active or inactive against *C. albicans*. This suggests that the introduction of a phenacyl moiety may have interfered with their antifungal activity. However, substituting the phenacyl moiety at the *para* position with a methoxy group, combined with the incorporation of a phenyl group into the core scaffold, restored antifungal activity as shown in compound **7f**. On the other hand, although the angular furobenzopyrone derivatives **3a** and **3b** demonstrated good antifungal activity against *C. albicans* comparable to amphotericin B, compounds **8a**-**8f** were inactive. This could indicate an unfavorable steric effect of the bulky group (phenyl ring), attached to the furan ring, with their antifungal activity. While tetrahydrobenzofurobenzopyrone **5a** and benzofurobenzopyrones **6a** and **6b** exhibited antifungal activity against *C. albicans*, compound **5b** was inactive (Table 2).

The antimicrobial results revealed that compounds **6a**, **6b**, **7a**, **7b**, and **8a**-**8f** were active against all tested strains of Gram-positive bacteria (*B. subtilis* and *S. aureus*) and Gram-negative bacteria (*E. coli*), suggesting a broad spectrum of antibacterial activity. Interestingly, all furobenzopyrone derivatives **8a**-**8f**, with phenyl substituent incorporated into the furan ring, retained their broad-spectrum antibacterial activity despite losing antifungal activity. Additionally, the results indicated that the molecular weight and lipophilicity of these derivatives were suitable for cellular induction, enabling them to exhibit antimicrobial activity intracellularly.

### 2.3. Photosensitizing Assay

The synthesized derivatives were evaluated for their photosensitizing activity against *B. subtilis*, expressed as minimum inhibitory concentrations (MIC, mg mL^−1^), using xanthotoxin as standard (Table 3 and Figure 4). The results revealed that benzopyrone derivatives **4a** and **7b** possessed stronger photosensitizing activity than xanthotoxin. Compound **7a** demonstrated comparable activity, whereas compounds **7c** and **7d** showed lower activity compared to xanthotoxin. Notably, benzopyrone derivatives **4a** and **7a**-**7c**, which have bulky groups (cyclohexyloxy or phenacyloxy) and small groups (R_1_ = Me), exhibited photosensitizing activity. In contrast, compounds **2a**, **2b**, **4b** and **7d** exhibited antimicrobial activity against *B. subtilis* without any photosensitizing activity after 20 min of irradiation.

Tetrahydrobenzofurobenzopyrones **5a** and **5b** and benzofurobenzopyrone derivative **6a** exhibited good photosensitizing activity comparable to xanthotoxin, while compound **6b** demonstrated similar activity. Among the furobenzopyrone series, compounds **8a**-**8e** showed the highest photosensitizing activity. On the other hand, compounds **3a** and **3b** exhibited antimicrobial activity against *B. subtilis* without any photosensitizing activity after 20 min of irradiation. Interestingly, all furobenzopyrone derivatives **8a**-**8f**, with a phenyl substituent incorporated into the furan ring, exhibited photosensitizing activity. The photosensitizing assay, combined with antimicrobial evaluation, demonstrated that derivatives containing aromatic tetracyclic-based scaffolds (**6a**, **6b**, **7a**-**7f**, and **8a**-**8f**) exhibited dual antimicrobial and photosensitizing activities.

### 2.4. Binding Mode Investigation Using Molecular Docking

To identify the possible mechanism of action of the designed benzopyrone-based derivatives, confirm their binding to the ATP binding site of GyrB, and gain insight into their molecular interactions, in silico molecular docking was conducted on the *N*-terminal domain of the *E. coli* DNA gyrase B subunit (GyrB24kDa). The most active compounds among the tested derivatives in microbiological and photosensitizing assay (**6a**, **6b**, **7a**-**7f**, and **8a**-**8f**) were selected for molecular docking into the ATP binding site of the *N*-terminal domain of gyrB subunit (PDB ID: 6YD9) [1], using the reference ligand (an ATP competitive agent; benzothiazole derivative II) as a standard. The molecular docking calculations were performed by AutoDock software 4.2.6, employing an empirical free energy force field and Lamarckian genetic algorithm (LGA) [35,36]. The reference ligand (benzothiazole derivative II) was re-docked into its active site (ATP binding site) to validate the molecular docking protocol (docking score = −8.45 kcal/mol and RMSD = 1.6786) before conducting the molecular docking of the tested compounds. The generated conformational clusters were visualized and analyzed by Discovery Studio 2021 software. The molecular docking results are summarized in Table 4.

The docking results revealed that the tested compounds (**6a**, **6b**, **7a**-**7f**, and **8a**-**8f**) and the reference ligand (**II**) shared similar binding interactions with the key amino acid residues within the ATP binding site of gyrase B subunit (Figure 5): hydrophobic pocket (Val43, Asn46, Val71, Ile94, Val120, Thr165, and Val167), hydrophobic surface (Glu50 and Arg76), and solvent exposed area (Ile78, Pro79, and Arg136). Additionally, the newly designed compounds exhibited strong hydrogen bonding interactions with Gly77 (rigid tetracyclic-based compounds **6a** and **6b** and flexible-linker containing compounds **7a**-**7f**) and Thr165 (rigid tricyclic-based derivatives, **8a**-**8f**). Moreover, the tricyclic angular furobenzopyrone derivatives (**8a**-**8f**) showed better docking scores compared to the rigid tetracyclic-based derivatives (**6a** and **6b**, Figure 5B), the flexible-linker-containing benzopyrone derivatives (**7a**-**7f**, Figure 5C), and the reference ligand **II** (Figure 5A).

The analysis of the generated conformational clusters of the rigid derivatives (**8a**-**8f**) revealed that the rigid and angular furan ring oriented the distal phenyl ring to fit into the valine-rich hydrophobic pocket, facilitating hydrophobic interactions with the key amino acid residues at this valine-rich region (Figure 5D). In addition, these compounds (**8a**-**f**) exhibited remarkably strong hydrogen bonding interactions with Gly77 and Thr165 (<2.3 Å for conformers with highest docking scores; **8d**-**8f**). The benzopyrone central scaffold, across the rigid series, was positioned within the ATP binding site to interact with the key amino acid residues at the ceiling of the binding site (Glu50 and Arg76), leaving the methyl group (**8a**-**8c**), and phenyl ring (**8d-8f**) anchored to the solvent-exposed area and exhibited a number of hydrophobic interactions with the key amino acid residues at that region (Ile78, Pro79, and Arg136). Although compounds **6a** and **6b** shared similar chemical fragments with compounds **8a** and **8b**, they did not show the key hydrophobic pattern with the key amino acid residues within the valine-rich hydrophobic pocket. The docking results suggest that the fused phenyl ring of compounds **6a** and **6b** could not fully access the hydrophobic pocket, in contrast to the distal phenyl ring of compounds **8a**-**8f** which was reflected in their virtual docking scores.

In general, the rigid tricyclic benzopyrone derivatives (**8a**-**8f**) showed docking conformations with coplanarity among the aromatic rings, allowing the key chemical moieties to orient themselves to effectively engage with the key regions within the ATP binding site through strong interactions (HB interaction between carbonyl group of tricyclic benzopyrone derivatives and Gly77 or Thr165 in addition to the hydrophobic interactions between distal phenyl ring and the valine-rich pocket) in a competitive manner with ATP molecules (Figure 5 and Figure 6).

The docking results indicated that, across methoxy-containing derivatives (**7c**, **7f**, **8c**, and **8f**), the hydrogen bond acceptor group (oxygen) failed to establish any H-bonding within the hydrophobic pocket. This suggests that the orientation of the distal phenyl ring in these derivatives prevents the methoxy group (OMe) from being close enough to the polar groups at the ATP binding site. Conversely, the docking results of the flexible linker-containing derivatives (**7a**-**7f**), as opposed to the rigid-based derivatives **8a**-**8f**, showed 90° torsion of the distal phenyl ring relative to the central benzopyrone scaffold. This loss of coplanarity between the rings of derivatives hinders the distal phenyl ring from fitting perfectly, thereby preventing stronger interactions with the key amino acid residues at the valine-rich hydrophobics pocket. These molecular docking results can explain the differences in the antimicrobial activities between compounds **7a**-**7f** and compounds **8a**-**8f**.

### 2.5. In Vitro DNA Gyrase Supercoiling Inhibitory Assay

Compounds **8a**, **8b**, and **8d** were selected among the most active series (based on antimicrobial evaluation, photosensitizing assay, and molecular docking) to be evaluated for their in vitro DNA gyrase supercoiling inhibitory activity (Table 5). The inhibitory assay was conducted using different concentrations of the tested derivatives (**8a**, **8b**, and **8d**) and standard drugs to calculate their IC_50_ values (5.0, 2.5, 1.25, 0.625, 0.312, 0.156, and 0.078 U/mL). The results revealed that the tested compounds displayed different IC_50_ values compared to the standard compounds. Compound **8b** (with methyl-substituted distal phenyl ring), showed weaker IC_50_ values (IC_50_ 12.30 μg mL^−1^ or 38.63 μM) compared to the other tested compounds (**8a** and **8d**) and the standards. Notably, compounds **8a** and **8d** (with unsubstituted distal phenyl ring) demonstrated potent IC_50_ values compared to the standard compounds. Additionally, compound **8d** (R_1_ = Ph) exhibited the most potent IC_50_ value compared to its methyl analogue (**8a**, R_1_ = Me).

Consistent with the molecular docking results, the rigid and planar tetracyclic-based scaffold plays a crucial role in competitively binding to the ATP binding site of the gyrase B *N*-terminal domain. Furthermore, the unsubstituted distal phenyl ring acts as an ideal binding motif within the valine-rich hydrophobic pocket. Introducing any additional substitution to this phenyl ring may hinder its full access to the hydrophobic pocket (**8a** in contrast to **8b**). Moreover, the phenyl ring, attached to the benzopyrone core scaffold facilitates multiple hydrophobic interactions with the key amino acid residues in the solvent-exposed area (**8d** in contrast to **8a**).

### 2.6. In Silico ADME Prediction

The In silico prediction of ADME profile and other related parameters was evaluated for the most active compound in the antimicrobial assay, photosensitizing activity, in silico molecular docking screening, and in vitro DNA gyrase supercoiling assay (compound **8d**) using the online-based SwissADME tool [37] using omeprazole as a reference drug (Table 6). The in silico prediction protocol based on Lipinski’s rule of five to assess the drug-likeness of chemical candidates; molecular weight (≤500 Dalton), number of rotatable bonds (≤10), number of H-bond acceptors (≤10), number of H-bond donors (≤5), molar reactivity between 40 and 130, topological polar surface area (≤140 Å^2^), and Log P*_o/w_* lipophilicity (≤4.15), Log S (estimated aqueous solubility, >−6) [38,39].

Compound **8d** showed a Log P value of 5.48 (ideal value ≤ 4.15) and a Log S (ESOL) value of −6.30 (ideal value > −6). These results indicate that compound **8d** is highly lipophilic and likely to have poor aqueous solubility. This suggests the need to introduce additional water-soluble polar groups into the designed compounds to enhance their polarity and improve water solubility.

In terms of predicted pharmacokinetic profiling, it is not surprising that compound **8d** exhibits high gastrointestinal (GI) absorption compared to the reference drug (omeprazole). Despite its high lipophilicity, it is predicted that compound **8d** cannot pass the blood-brain barrier (BBB). These results suggest that the designed derivatives may not induce any central off-target effects. The drug-likeness analysis demonstrated that compound **8d** exhibits strong potential for drug-like characteristics, as it did not violate any of Lipinski’s rules and met the criteria for drug-likeness, achieving a bioavailability score of 0.55.

## 3. Materials and Methods

### 3.1. Chemistry

#### 3.1.1. General

Melting points were measured using an open capillary tube method with Stuart SMP10 melting point instrument and are uncorrected. Microanalysis was performed at the Regional Center for Mycology and Biotechnology, Al-Azhar University. The ^1^H NMR spectra were recorded on a Bruker (400 MHz) NMR spectrometer in chloroform (CDCl_3_) or dimethylsulphoxide (DMSO-*d*_6_). Chemical Shifts are reported in *δ* as parts per million (ppm) downfield from tetramethylsilane (TMS) as an internal standard, and *J* values were computed in Hz. Mass spectra were recorded as EI at 70 eV using Hewlett Packard Varian (Varian, Polo, Illinois, USA) and Shimadzu Gas Chromatograph Mass spectrometer-QP 1000 EX. TLC was performed using Macherey-Nagel AlugramSil G/UV254 silica gel plates with fluorescent indicator UV254, and the elution system was chloroform/methanol 98:2. The spots were detected at λ = 254 nm using a UV VilberLourmat 77202 (Vilber, Marne La Vallee, France).

#### 3.1.2. General Procedure for Synthesis of 6-Ethyl-7-(3-oxobutan-2-yloxy)-4-substituted-2H-1-benzopyran-2-ones (**2a**,**b**)

A solution of compound **1a** or **1b** (0.01 mol) and 3-chlorobutan-2-one (1.06 g, 0.01 mol) in acetone (30 mL) and anhydrous potassium carbonate (2.76 g, 0.02 mol) was refluxed for 20 h. The solution was concentrated to dryness, and the formed residue was washed with water, dried, and finally crystallized from ethanol to give the titled products **2a** or **2b**.

6-Ethyl-4-methyl-7-(3-oxobutan-2-yloxy)-2H-1-benzopyran-2-one (**2a**)

Yield 85%; mp 95–97 °C; ^1^H NMR (400 MHZ, DMSO-*d*_6_) *δ* 1.21 (t, 3H, CH_2_CH_3_), 1.48 (d, 3H, *J* 8.0 Hz, OCHCH_3_), 2.21 (s, 3H, COCH_3_), 2.40 (s, 3H, CH_3_), 2.68 (q, 2H, CH_2_CH_3_), 5.16 (q, 1H, OCHCH_3_), 6.19 (s, 1H, Ar-H), 6.84 (s, 1H, Ar-H), 7.53 (s, 1H, Ar-H); MS (EI) *m/z*, calcd. for C_16_H_18_O_4_ (M^+^): 274.12, found: 274.15; Anal. Calc.: C, 70.06; H, 6.61, found: C, 70.22; H, 6.69.

6-Ethyl-7-(3-oxobutan-2-yloxy)-4-phenyl-2H-1-benzopyran-2-one (**2b**)

Yield 93%; mp 100–102 °C; ^1^H NMR (400 MHZ, DMSO-*d*_6_) *δ* 1.09 (t, 3H, CH_2_CH_3_), 1.50 (d, 3H, *J* 8.0 Hz, OCHCH_3_), 2.24 (s, 3H, COCH_3_), 2.59 (q, 2H, CH_2_CH_3_), 5.21 (q, 1H, OCHCH_3_), 6.21 (s,1H, H-3), 6.97 (s,1H, Ar-H), 7.20 (s,1H, Ar-H), 7.52–7.61 (m, 5H, Ph-4); MS (EI) *m/z*, calcd. for C_21_H_20_O_4_ (M^+^): 336.14, found: 336.05; Anal. Calc.: C, 74.98; H, 5.99, found: C, 75.19; H, 6.05.

#### 3.1.3. General Procedure for Synthesis of 2,3-Dimethyl-9-ethyl-7-substituted-5H-furo[2,3-h]benzopyran-5-ones (**3a**,**b**)

A solution of compound **2a** or **2b** (0.002 mol) in absolute ethanol (15 mL) was treated with 1N NaOH (15 mL) and refluxed for 4–12 h (TLC monitored). The reaction was left to be concentrated overnight at room temperature. The residue was collected, acidified with 1N H_2_SO_4_, filtered, and dried. Finally, the crude product was crystallized from ethanol/petroleum ether to give the titled products **3a,b**.

9-Ethyl-2,3,7-trimethyl-5*H*-furo[2,3-h]benzopyran-5-one (**3a**)

Yield 76%; Mp 199–201 °C; ^1^H NMR (400 MHZ, DMSO-*d*_6_) *δ* 1.18 (t, 3H, CH_2_CH_3_), 2.03 (s, 3H, CH_3_), 2.13 (s, 3H, CH_3_), 2.22 (s, 3H, CH_3_), 2.67 (q, 2H, CH_2_CH_3_), 6.73 (s, 1H, Ar-H), 7.37 (s, 1H, Ar-H); MS (EI) *m/z*, calcd. for C_16_H_16_O_3_ (M^+^): 256.11, found: 256.10; Anal. Calc.: C, 74.98; H, 6.29, found: C, 75.13; H, 6.38.

2,3-Dimethyl-9-ethyl-7-phenyl-5H-furo[2,3-h]benzopyran-5-one (**3b**)

Yield 63%; mp 90–92 °C; ^1^H NMR (400 MHZ, CDCl_3_) *δ* 1.26 (t, 3H, CH_2_CH_3_), 2.44 (s, 3H, CH_3_), 2.50 (s, 3H, CH_3_), 2.84 (q, 2H, CH_2_CH_3_), 6.30 (s, 1H, Ar-H), 7.06 (s, 1H, Ar-H), 7.28–7.58 (m, 5H, Ph-7); MS (EI) *m/z*, calcd. for C_21_H_18_O_3_ (M^+^): 318.13, found: 317.90; Anal. Calc.: C, 79.22; H, 5.70, found: C, 79.47; H, 5.76.

#### 3.1.4. General Procedure for Synthesis of 6-Ethyl-7-(2-oxocyclohexyloxy)-4-substituted-2H-1-benzopyran-2-ones (**4a**,**b**)

A solution of compound **1a** or **1b** (0.01 mol) in acetone (30 mL) was treated with anhydrous potassium carbonate (2.76 g, 0.02 mol) and 2-chlorocyclohexanone (1.32 g, 0.01 mol). The reaction mixture was refluxed for 18–26 h and then filtered, and the obtained residue was rinsed with acetone. The filtrates and washings were combined and evaporated under reduced pressure. The obtained product was crystallized from ethanol/petroleum ether to give the titled products **4a** or **4b**.

6-Ethyl-4-methyl-7-(2-oxocyclohexyloxy)-2H-1-benzopyran-2-one (**4a**)

Yield 83%; mp 160–162 °C; ^1^H NMR (400 MHZ, DMSO-*d*_6_) *δ* 1.18 (t, 3H, CH_2_CH_3_), 1.79–1.89 (m, 2H, Ch-H*), 2.00–2.05 (m, 2H, Ch-H*), 2.30–2.33 (m, 2H, Ch-H*), 2.40 (s, 3H, CH_3_), 2.51 (t, 2H, Ch-H*), 2.65 (q, 2H, CH_2_CH_3_), 5.22 (t, 1H, Ch-H*), 6.16 (s, 1H, Ar-H), 6.89 (s, 1H, Ar-H), 7.5 (s, 1H, Ar-H); MS (EI) *m/z*, calcd. for C_18_H_20_O_4_ (M^+^): 300.14, found: 300.16; Anal. Calc.: C, 71.98; H, 6.71, found: C, 72.14; H, 6.74; *Ch-H: Cyclohexanone hydrogens.

6-Ethyl-7-(2-oxocyclohexyloxy)-4-phenyl-2H-1-benzopyran-2-one (**4b**)

Yield 86%; mp 150–152 °C; ^1^H NMR (400 MHZ, DMSO-*d*_6_) *δ* 1.08 (t, 3H, CH_2_CH_3_), 1.54–1.67 (m, 2H, Ch-H*), 1.80–1.90 (m, 2H, Ch-H*), 2.02–2.06 (m, 2H, Ch-H*), 2.31–2.41 (m, 2H, Ch-H*), 2.56 (q, 2H, CH_2_CH_3_), 5.27 (t, 1H, Ch-H*), 6.19 (s,1H, Ar-H), 7.02 (s,1H, Ar-H), 7.17 (s,1H, Ar-H), 7.46–7.61 (m, 5-H, Ph-4); MS (EI) *m/z*, calcd. for C_23_H_22_O_4_ (M^+^): 362.15, found: 362.00; Anal. Calc.: C, 76.22; H, 6.12, found: C, 76.39; H, 6.26; *Ch-H: Cyclohexanone hydrogens.

#### 3.1.5. General Procedure for Synthesis of 6-Ethyl-4-substituted-8,9,10,11-tetrahydro-2H-benzofuro[2,3-h]benzopyran-2-ones (**5a**,**b**)

A suspension of compound **4a** or **4b** (0.002 mol) in absolute ethanol (15 mL) was treated with 1N NaOH (15 mL) and refluxed for 28 h till complete dissolution of the starting compound. The solution was filtered while hot and then cooled. The filtrate was acidified with 1N H_2_SO_4_. The formed precipitate was filtered, dried, and crystallized from ethanol/petroleum ether to give the titled product **5a** or **5b**.

6-Ethyl-4-methyl-8,9,10,11-tetrahydro-2H-benzofuro[2,3-h]benzopyran-2-one (**5a**)

Yield 71%; mp 150–152 °C; ^1^H NMR (400 MHZ, CDCl_3_) *δ* 1.37 (t, 3H, CH_2_CH_3_), 1.84–1.99 (m, 4H, Thb-H*), 2.42 (s, 3H, CH_3_), 2.77–3.01 (m, 4H, Thb-H*), 2.94 (q, 2H, CH_2_CH_3_), 6.22 (s, 1H, Ar-H), 7.18 (s, 1H, Ar-H); MS (EI) *m/z*, calcd. for C_18_H_18_O_3_ (M^+^): 282.13, found: 282.10; Anal. Calc.: C, 76.57; H, 6.43, found: C, 76.80; H, 6.49; *Thb-H: Tetrahydrobenzo hydrogens.

6-Ethyl-4-phenyl-8,9,10,11-tetrahydro-2H-benzofuro[2,3-h]benzopyran-2-one (**5b**)

Yield 56%; mp 108–110 °C; ^1^H NMR (400 MHZ, DMSO-*d*_6_) *δ* 1.00–1.03 (m, 4H, Thb-H*), 1.15 (t, 3H, CH_2_CH_3_), 2.71 (q, 2H, CH_2_CH_3_), 3.43–3.75 (m, 4H, Thb-H*), 6.28 (s, 1H, Ar-H), 6.99 (s, 1H, Ar-H), 7.51–7.60 (m, 5H, Ph-4); Anal. Calc. for C_23_H_20_O_3_ (344.40): C, 80.21; H, 5.85, found: C, 80.34; H, 5.91; *Thb-H: Tetrahydrobenzo hydrogens.

#### 3.1.6. General Procedure for Synthesis of 6-Ethyl-4-substituted-2H-benzofuro[2,3-h]benzopyran-2-ones (**6a**,**b**)

A mixture of compound **5a** or **5b** (0.003 mol) and 2,3-dichloro-5,6-dicyano-1,4-benzo-quinone (DDQ) (0.5 g) in benzene (30 mL) was refluxed for 20 h. The reaction mixture was filtered while hot and then concentrated to dryness. The obtained residue was collected and crystallized from ethanol to give the titled products **6a** or **6b**.

6-Ethyl-4-methyl-2H-benzofuro[2,3-h]benzopyran-2-one (**6a**)

Yield 40%; mp 120–122 °C; ^1^H NMR (400 MHZ, CDCl_3_) *δ* 1.38 (t, 3H, CH_2_CH_3_), 2.53 (s, 3H, CH_3_), 3.04 (q, 2H, CH_2_CH_3_), 6.30 (s, 1H, Ar-H), 7.37 (s,1H, Ar-H), 7.42 (t, 1H, Ar-H), 7.53 (t, 1H, Ar-H), 7.62 (d, 1H, *J* 8.0 Hz, Ar-H), 8.39 (d, 1H, *J* 7.4 Hz, Ar-H); MS (EI) *m/z*, calcd. for C_18_H_14_O_3_ (M^+^): 278.09, found: 277.90; Anal. Calc.: C, 77.68; H, 5.07. Found: C, 77.92; H, 5.16.

6-Ethyl-4-phenyl-2H-benzofuro[2,3-h]benzopyran-2-one (**6b**)

Yield 41%; mp 120–122 °C; ^1^H NMR (400 MHZ, DMSO-*d*_6_) *δ* 1.18 (t, 3H, CH_2_CH_3_), 2.77 (q, 2H, CH_2_CH_3_), 6.32 (s,1H, Ar-H), 7.02 (s,1H, Ar-H), 7.50- 7.66 (m, 9H, 5Ph-H, 4Ar-H); MS (EI) *m/z*, calcd. for C_23_H_16_O_3_ (M^+^): 340.11, found: 340.00; Anal. Calc.: C, 81.16; H, 4.74, found: C, 81.39; H, 4.72.

#### 3.1.7. General Procedure for Synthesis of 6-Ethyl-4-substituted-7-(un)substituted phenacyloxy-2H-1-benzopyran-2-ones (**7a**-**f**)

An appropriate 4-substituted phenacyl bromide (0.015 mol) was added to a solution of hydroxyl compound **1a** or **1b** (0.01 mol) in acetone (50 mL) while stirring in the presence of anhydrous potassium carbonate (2.76 g, 0.02 mol). The mixture was refluxed for 14 h (TLC monitored). Acetone was distilled off and the residue was rinsed with cold water, filtered, and dried. Finally, the crude residue was crystallized from ethanol to give the titled products **7a**-**7f**.

6-Ethyl-4-methyl-7-phenacyloxy-2H-1-benzopyran-2-one (**7a**)

Yield 96%; mp 165–166 °C; ^1^H NMR (400 MHZ, DMSO-*d*_6_) *δ* 1.23 (t, 3H, CH_2_CH_3_), 2.41 (s, 3H, CH_3_), 2.73 (q, 2H, CH_2_CH_3_), 5.77 (s, 2H, OCH_2_CO), 6.19 (s, 1H, Ar-H), 7.06 (s, 1H, Ar-H), 7.55–7.73 (m, 4H, 1Ar-H and 3H, Ph-H), 8.04 (d, 2H, *J* 8.0 Hz, Ph-H); MS (EI) *m/z*, calcd. for C_20_H_18_O_4_ (M^+^): 322.12, found: 322.14. Anal. Calc.: C, 74.52; H, 5.63, found: C, 74.78; H, 5.69.

6-Ethyl-4-methyl-7-(4-methylphenacyloxy)-2H-1-benzopyran-2-one (**7b**)

Yield 95%; mp 136–139 °C; ^1^H NMR (400 MHZ, DMSO-*d*_6_) *δ* 1.21 (t, 3H, CH_2_CH_3_), 2.19 (s, 3H, CH_3_), 2.40 (s, 3H, CH_3_), 2.67 (q, 2H, CH_2_CH_3_), 5.72 (s, 2H, OCH_2_CO), 6.18 (s, 1H, Ar-H), 6.78–8.06 (m, 6H, 2Ar-H and 4H, Ph-H); MS (EI) *m/z*, calcd. for C_21_H_20_O_4_ (M^+^): 336.14, found: 336.79; Anal. Calc.: C, 74.98; H, 5.99, found: C, 75.14; H, 6.12.

6-Ethyl-4-methyl-7-(4-methoxyphenacyloxy)-2H-1-benzopyran-2-one (**7c**)

Yield 93%; mp 162–164 °C; ^1^H NMR (400 MHZ, DMSO-*d*_6_) *δ* 1.22 (t, 3H, CH_2_CH_3_), 2.41 (s, 1H, CH_3_), 2.72 (q, 2H, CH_2_CH_3_), 3.87 (s, 3H, OCH_3_), 5.69 (s, 2H, OCH_2_CO), 6.18 (s, 1H, Ar-H), 7.02 (s, 1H, Ar-H), 7.10 (d, 2H, *J* 8.0 Hz, Ph-H), 7.54 (s, 1H, Ar-H), 8.02 (d, 2H, *J* 8.0 Hz, Ph-H); MS (EI) *m/z*, calcd. for C_21_H_20_O_5_ (M^+^): 352.13, found: 352.05; Anal. Calc.: C, 71.58; H, 5.72, found: C, 71.81; H, 5.79.

6-Ethyl-7-phenacyloxy-4-phenyl-2H-1-benzopyran-2-one (**7d**)

Yield 97%; mp 153–155 °C; ^1^H NMR (400 MHZ, DMSO-*d*_6_) *δ* 1.12 (t, 3H, CH_2_CH_3_), 2.64 (q, 2H, CH_2_CH_3_), 5.82 (s, 2H, OCH_2_CO), 6.21 (s, 1H, Ar-H), 7.20 (d, 2H, *J* 8.0 Hz, Ph-H), 7.51–7.80 (m, 8H, 2Ar-H and 6H, Ph-H), 8.06 (d, 2H, *J* 8.0 Hz, Ph-H); MS (EI) *m/z*, calcd. for C_25_H_20_O_4_ (M^+^): 384.14, found: 384.09; Anal. Calc.: C, 78.11; H, 5.24, found: C, 78.34; H, 5.31.

6-Ethyl-7-(4-methylphenacyloxy)-4-phenyl-2H-1-benzopyran-2-one (**7e**)

Yield 95%; mp 225–226 °C; ^1^H NMR (400 MHZ, DMSO-*d*_6_) *δ* 1.17 (t, 3H, CH_2_CH_3_), 2.47 (s, 3H, CH_3_) 2.69 (q, 2H, CH_2_CH_3_), 5.83 (s, 2H, OCH_2_CO), 6.27 (s, 1H, Ar-H), 7.22 (s,1H, Ar-H), 7.27 (s, 1H, Ar-H), 7.46 (d, 2H, *J* 8.0 Hz, Ph-H), 7.59–7.66 (m, 5H, Ph-H), 8.01 (d, 2H, *J* 8.0 Hz, Ph-H); MS (EI) *m/z*, calcd. for C_26_H_22_O_4_ (M^+^): 398.15, found: 398.17; Anal. Calc.: C, 78.37; H, 5.57, found: C, 78.51; H, 5.61.

6-Ethyl-7-(4-methoxyphenacyloxy)-4-phenyl-2H-1-benzopyran-2-one (**7f**)

Yield 96%; mp 183–185 °C; ^1^H NMR (400 MHZ, DMSO-*d*_6_) *δ* 1.12 (t, 3H, CH_2_CH_3_), 2.63 (q, 2H, CH_2_CH_3_), 3.88 (s, 1H, OCH_3_), 5.74 (s, 2H, OCH_2_CO), 6.21(s, 1H, Ar-H), 7.14 (d, 2H, *J* 8.0 Hz, Ph-H), 7.21 (s, 1H, Ar-H), 7.54–7.60 (m, 6H, 1Ar-H and 5H, Ph-H), 8.04 (d, 2H, *J* 8.0 Hz, Ph-H); MS (EI) *m/z*, calcd. for C_26_H_22_O_5_ (M^+^): 414.15, found: 414.18; Anal. Calc.: C, 75.35; H, 5.35, found: C, 75.49; H, 5.32.

#### 3.1.8. General Procedure for Synthesis of 9-Ethyl-7-substituted-3-(un)substituted phenyl-5H-furo[2,3-h]benzopyran-5-ones (**8a**-**8f**)

Compound **7a**-**7f** (0.003 mol) was added to sodium ethoxide solution (0.07 g, 0.003 mol sodium metal in 50 mL ethanol) and the mixture was refluxed for 2 h. The solution was concentrated and then acidified with cold 10% HCl. The formed precipitate was filtered, washed with water, and crystallized from ethanol to give the titled products **8a**-**8f**.

9-Ethyl-7-methyl-3-phenyl-5H-furo[2,3-h]benzopyran-5-one (**8a**)

Yield 40%; mp 103–105 °C; ^1^H NMR (400 MHZ, DMSO-*d*_6_) *δ* 1.34 (t, 3H, CH_2_CH_3_), 2.44 (s, 3H, CH_3_), 2.95 (q, 3H, CH_2_CH_3_), 6.32 (s, 1H, Ar-H), 7.56 (s, 1H, Ar-H), 7.42–7.56 (m, 3H, Ph-H), 7.72 (d, 2H, *J* 8.0 Hz, Ph-H), 8.28 (s, 1H, Ar-H); MS (EI) *m/z*, calcd. for C_20_H_16_O_3_ (M^+^): 304.11, found: 304.05; Anal. Calc.: C, 78.93; H, 5.30, found: C, 79.14; H, 5.28.

9-Ethyl-7-methyl-3-(4-methylphenyl)-5H-furo[2,3-h]benzopyran-5-one (**8b**)

Yield 42%; mp 149–151 °C; ^1^H NMR (400 MHZ, DMSO-*d*_6_) *δ* 1.16 (t, 3H, CH_2_CH_3_), 2.33 (s, 3H, CH_3_), 2.40 (s, 3H, CH_3_), 2.67 (q, 3H, CH_2_CH_3_), 6.23 (s, 1H, Ar-H), 6.69–7.67 (m, 5H, 1Ar-H and 4H, Ph-H), 8.25 (s, 1H, Ar-H); MS (EI) *m/z*, calcd. for C_21_H_18_O_3_ (M^+^): 318.13, found: 318.09; Anal. Calc.: C, 79.22; H, 5.70, found: C, 79.41; H, 5.78.

9-Ethyl-3-(4-methoxyphenyl)-7-methyl-5H-furo[2,3-h]benzopyran-5-one (**8c**)

Yield 44%; mp 110–111 °C; ^1^H NMR (400 MHZ, DMSO-*d*_6_) *δ* 1.22 (t, 3H, CH_2_CH_3_), 2.41 (s, 3H, CH_3_), 2.71 (q, 2H, CH_2_CH_3_), 3.87 (s, 3H, OCH_3_), 6.18 (s, 1H, Ar-H), 7.02 (s, 1H, Ar-H), 7.10 (d, 2H, *J* 8.0 Hz, Ph-H), 8.02 (d, 2H, *J* 8.0 Hz, Ph-H), 8.22 (s, 1H, Ar-H); MS (EI) *m/z*, calcd. for C_21_H_18_O_4_ (M^+^): 334.12, found: 334.13; Anal. Calc.: C, 75.43; H, 5.43, found: C, 75.68; H, 5.47.

3,7-Diphenyl-9-ethyl-5H-furo[2,3-h]benzopyran-5-one (**8d**)

Yield 94%; mp 178–180 °C; ^1^H NMR (400 MHZ, DMSO-*d*_6_) *δ* 1.23 (t, 3H, CH_2_CH_3_), 2.88 (q, 2H, CH_2_CH_3_), 6.35 (s, 1H, Ar-H), 7.21 (s, 1H, Ar-H), 7.44–7.61 (m, 8H, Ph-H), 7.76 (d, 2H, *J* 8.0 Hz, Ph-H), 8.34 (s, 1H, Ar-H); MS (EI) *m/z*, calcd. for C_25_H_18_O_3_ (M^+^): 366.13, found: 366.21; Anal. Calc.: C, 81.95; H, 4.95, found: C, 82.08; H, 5.01.

9-Ethyl-3-(4-methylphenyl)-7-phenyl-5H-furo[2,3-h]benzopyran-5-one (**8e**)

Yield 94%; mp 80–81 °C; ^1^H NMR (400 MHZ, DMSO-*d*_6_) *δ* 1.21 (t, 3H, CH_2_CH_3_), 2.31 (s, 3H, CH_3_), 2.78 (q, 3H, CH_2_CH_3_), 6.11(s, 1H, Ar-H), 7.33–7.69 (m, 10H, 1Ar-H and 9H, Ph-H), 8.27 (s, 1H, Ar-H); MS (EI) *m/z*, calcd. for C_26_H_20_O_3_ (M^+^): 380.14, found:380.17; Anal. Calc.: C, 82.08; H, 5.30, found: C, 82.17; H, 5.34.

9-Ethyl-3-(4-methoxyphenyl)-7-phenyl-5H-furo[2,3-h]benzopyran-5-one (**8f**)

Yield 90%; mp 150–153 °C; ^1^H NMR (400 MHZ, DMSO-*d*_6_) *δ* 1.12 (t, 3H, CH_2_CH_3_), 2.63 (q, 2H, CH_2_CH_3_), 3.88 (s, 3H, OCH_3_), 6.35 (s, 1H, Ar-H), 7.11 (d, 2H, *J* 8.6 Hz, Ph-H), 7.21 (d, 2H, Ph-H), 7.30 (s, 1H, Ar-H), 7.51–7.70 (m, 3H, Ph-H), 8.03 (d, 2H, *J* 8.0 Hz, Ph-H), 8.26 (s, 1H, Ar-H); MS (EI) *m/z*, calcd. for C_26_H_20_O_4_ (M^+^): 396.14, found: 396.22; Anal. Calc.: C, 78.77; H, 5.09, found: C, 78.84; H, 5.15.

### 3.2. Antimicrobial Assay

#### 3.2.1. General

The antimicrobial activity of all newly synthesized compounds was determined using the agar cup-plate method using the Hi-Media agar medium [40]. Xanthotoxin, tobramycin, and amphotericin B were used as reference drugs. All newly synthesized compounds were tested against two strains of Gram-positive bacteria (*Bacillus subtilis* ATCC 14579 and *Staphylococcus aureus* ATCC 25923), Gram-negative bacteria (*Escherichia coli* ATCC 25922), and yeast (*Candida albicans* ATCC 10231).

#### 3.2.2. Pre-Experimental Preparations

The nutrient agar medium consists of 0.3% of beef extract, 0.5% of peptone, 0.1% dipotassium hydrogen phosphate, and 1.5% agar. Broth cultures of the organisms were prepared by incubating slant agar seeded with the organisms overnight.

#### 3.2.3. Cup-Plate Method

The prepared broth culture (0.02 mL) was carefully added to sterile Petri dishes. Then, 10 mL of the liquefied nutrient agar medium was poured into each dish, mixed uniformly, and allowed to solidify. Using a sterilized cork borer, cups (9 mm) were scooped out of the agar media contained in the Petri dishes. Each test compound (50 mg) was dissolved in 1 mL DMF (50 mg mL^−1^), which was used as a stock solution to carry out the twofold dilution technique. A fixed volume (0.1 mL) of the tested compounds dissolved in dimethylformamide (DMF) was added. Each plate contained a cup filled with DMF to disregard the effect of solvent. Additionally, two other cups were filled with tobramycin and xanthotoxin being used as reference drugs. The plates were incubated overnight to examine the antimicrobial activity. Zones of inhibition obtained by each test compound were measured in millimeters and the results are recorded in Table 2.

#### 3.2.4. Statistical Analysis

To evaluate the significance of the difference in the results, each experiment was repeated at least three times, the data were represented as mean ± standard deviation, compared by Student’s t-test, and the p-values less than 0.05 were considered significantly different. GraphPad Prism-version 5 (GraphPad Software, Inc., La Jolla, CA, USA) was used for statistical analysis.

### 3.3. Photosensitizing Assay

The photosensitizing activity was determined using the agar cup-plate method (described in Section 2.3) with Hi-Media agar medium. The antibacterial activity against *Bacillus subtilis* ATCC 14579 was measured before and after exposure to UV lamp (365 nm). The diameters of the inhibition zones were measured in mm and the results were compared with xanthotoxin. The tested organism used was *Bacillus subtilis*.

For the photosensitizing activity, xanthotoxin was used as a reference drug and DMF as a negative control. The minimum inhibitory concentrations (*MIC*) were determined using the agar cup-plate method. Each test compound (50 mg) was dissolved in 1 mL DMF (50 mg mL^−1^), which was used as a stock solution to carry out the twofold dilution technique. The sample size for all the compounds was adjusted to 0.1 mL. The solution of the test compound (0.1 mL) was added to the cups. The plates were divided into test plates and duplicate plates. Before irradiation, the test plates were incubated in the dark at 37 °C for 3 h to permit the diffusion of the tested compounds through the agar layer. The duplicate plates were incubated overnight and used as a control for the determination of the antibacterial activity. The covers were removed from the test plates, and the dishes were exposed to UV lamp (365 nm) for 20 min. After irradiation, the plates were left overnight in the dark at 37 °C. MIC, the least concentration preventing visible bacterial growth, was determined in both the control and test, indicating the antibacterial and photosensitizing activity of the tested compounds, respectively.

To evaluate the significance of the difference in the results, each experiment was repeated at least three times. The data were represented as mean ± standard deviation and compared by Student’s t-test, and the p-values less than 0.05 were considered significantly different. GraphPad Prism-version 5 (GraphPad Software, Inc., La Jolla, CA, USA) was used to perform statistical analysis of the results. Results are recorded in Table 3.

### 3.4. In Silico Molecular Docking

The molecular docking of the designed derivatives (**6a**, **6b**, **7a**-**7f**, and **8a**-**8f**) was performed using Molecular Graphics Laboratory (MGL) Tools software suite 1.5.7 (Sanner lab, Centre for Computational Structural Biology, Scripps Research Institute). The molecular docking protocol was conducted using the following steps:

#### 3.4.1. Ligand Preparation

The chemical structures of the tested compounds (**6a**, **6b**, **7a**-**7f**, and **8a**-**8f**) were built using MarvinSketch 22.11 software and optimized by Discovery Studio 2021 software using the Dreiding-like force field [41]. Gasteiger charges were applied to assign partial charges to the atoms within the tested compounds using AutoDock Tools 1.5.6.

#### 3.4.2. Protein Preparation

The X-ray structure of the *N*-terminal of *E. coli* DNA gyrase B (PDB ID: 6YD9) was downloaded from the RCSB protein databank [1]. Reference ligands, water molecules, and any additional chains were removed, keeping the *N*-terminal domain only. The hydrogen atoms and Asn/Gln/His flips were assigned using Molprobity [42]. The Autodock4 force field (AD4) and Gasteiger charges were assigned to the protein atoms using AutoDock Tools 1.5.6. This force field uses a semi-empirical free energy force field to evaluate conformations during molecular docking. The protein was saved in an extended PDB format (PDBQT), which includes atomic partial charges, atom types, and information on the torsional degrees of freedom.

#### 3.4.3. Molecular Docking Protocol Validation

Molecular docking calculations were performed using AutoDock4 using ten runs of genetic algorithm (GA) at the ATP binding site coordinates (GyrB24kDa; x = 10.27, y = −4.32, z = 7.44). The docking protocol was validated by conducting initial docking experiments (pre-docking) for the reference ligand (**II**) and calculating the RMSD value (RMSD = 1.6876). The molecular docking of the designed compounds (**6a**, **6b**, **7a**-**7f**, and **8a**-**8f**) was conducted, and the docking poses with the highest docking scores were identified and explored.

#### 3.4.4. Molecular Docking Analysis

The docking poses of the tested compounds were visualized and analyzed using Discovery Studio software 2021 to identify the potential binding interactions between the docked ligands and the key amino acid residues at the ATP binding site of the DNA gyrase B subunit.

### 3.5. In Vitro DNA Gyrase Supercoiling Inhibitory Assay

Relaxed pBR322 DNA and *E.coli* DNA gyrase assay kits were obtained from John Innes Enterprises Ltd. (John Innes Centre and Norwich Research Park, Colney, Norwich, United Kingdom), courtesy of A. Maxwell. The supercoiling assay, based on DNA triplex, formation [43] was performed on the black streptavidin-coated 96-well microtiter plates (Thermo Scientific Pierce). Serial dilutions of the tested compounds and standard drugs were prepared, starting from 5.0 U/mL (5.0, 2.5, 1.25, 0.625, 0.312, 0.156, and 0.078 U/mL). The plates were first rehydrated with the supplied wash buffer [20 mM Tris-HCl (pH 7.6), 137 mM NaCl 0.01% (*w*/*v*), BSA 0.05% (*v*/*v*) Tween 20] and oligonucleotide was immobilized onto the wells. The excess oligonucleotide was then washed off, and the enzyme assay was carried out in the wells.

The final reaction volume of 30 μL in buffer [35 mM Tris × HCl (pH 7.5), 24 mM KCl, 4 mM MgCl_2_, 2 mM DTT (dithiothreitol), 1.8 mM spermidine, 1 mM ATP, 6.5% (*w*/*v*) glycerol, 0.1 mg/mL albumin] contained 1.5 U of gyrase from *E. coli*, 0.75 μg of relaxed pNO1 plasmid, and 3 μL of inhibitor solution in 10% DMSO and 0.008% Tween 20. Reactions were incubated for 30 min at 37 °C and after the addition of the TF buffer [50 mM NaOAc (pH 5.0), 50 mM NaCl, and 50 mM MgCl_2_], which terminated the enzymatic reaction for another 30 min at rt to allow the triplex formation. The unbound plasmid was then washed off using TF buffer, and a solution of SybrGOLD stain in T10 buffer [10 mM Tris × HCl (pH 8.0) and 1 mM EDTA] was added. After mixing, the fluorescence (excitation: 495 nm, emission: 537 nm) was measured using BioTek Synergy HT microplate reader. Ciprofloxacin and novobiocin were used as reference drugs. The inhibitory effects of the most potent compounds were examined and expressed as the concentration of the tested compound at which the DNA gyrase supercoiling activity was inhibited by 50% (IC_50_). The IC_50_ for the selected compounds and reference drugs are recorded in Table 5. GraphPad Prism-version 5 (GraphPad Software, Inc., La Jolla, CA, USA) was used to determine the IC_50_ values of both the tested compounds and standard drugs across the different concentrations.

### 3.6. In Silico ADME Prediction

The in silico prediction of the ADME profile (physicochemical properties, lipophilicity, water solubility, pharmacokinetic, and drug-likeness) was conducted for the most active compound (**8d**) using omeprazole as a standard. The ADME prediction was estimated according to Lipinski’s rule using the online-based SwissADME tool [37].

## 4. Conclusions

A series of benzopyrone-based derivatives was designed and synthesized to evaluate their antibacterial and photosensitizing activities. Most of the compounds exhibited different degrees of antimicrobial activity against *B. subtilis*, *S. aureus*, and *E. coli*. However, most of the tested compounds did not exhibit inhibition to *C. albicans* strain. Along with the antimicrobial assay, compounds **6a**, **6b**, **7a**-**7f**, and **8a**-**8f** showed promising photosensitizing activity against *B. subtilis* strain. Additional in silico and in vitro investigations were conducted to understand the possible mechanism of action and identify the potential binding modes of these compounds into the ATP binding site of the *N*-terminal domain of the DNA gyrase B subunit.

*In silico* molecular docking was carried out to the most active compounds (**6a**, **6b**, **7a**-**7f**, and **8a**-**8f**). The molecular docking results revealed that all of these compounds formed strong hydrogen bonds with Gly77 or Thr165. Moreover, the results showed that the coplanarity of the designed derivatives is crucial for full access of the distal phenyl ring to the valine-rich hydrophobic pocket. Compounds **8a**-**f** demonstrated the required coplanarity between the distal phenyl ring and the tetracyclic-based furobenzopyrone core scaffold, resulting in conformers with high docking scores into the ATP binding site of DNA gyrase B. However, compounds **6a** and **6b** (the rigid tetracyclic-based derivatives with coplanar structures) could not exhibit the same binding patterns as their analogues (compounds **8a**-**8f**). It is suggested that the fused phenyl ring in the furan moiety could not fully access the valine-rich hydrophobic pocket.

On the other hand, the flexible-spacer-containing benzopyrones (compounds **7a**-**7f**) did not show the same binding pattern at the valine-rich hydrophobic pocket as compounds **8a**-**8f**. The flexibility in these derivatives appears to allow the distal phenyl ring to twist and rotate, resulting in a loss of the necessary coplanarity with the core benzopyrone scaffold. Across the tested compounds, substitution at the distal phenyl ring is not favored, as it may hinder the full access of this hydrophobic binding moiety into the valine-rich pocket. This suggests that this hydrophobic pocket is small enough to accomodate any additional groups.

The in vitro DNA gyrase supercoiling inhibitory assay was performed to a selected group of the rigid tricyclic-based furobenzopyrone derivatives (**8a**,**8b**,**8d**). Compound **8d** showed the most potent IC_50_ value compared to the standard compounds (novobiocin and ciprofloxacin). The in vitro results suggested that the decoration of benzopyrone scaffold with phenyl substitution is favored than the methyl substitution. It is suggested that the phenyl ring provided additional hydrophobic interactions with the key amino acid residues at the solvent-exposed area of the DNA gyrase B domain.

Additionally, in silico ADME profiling prediction was conducted to the most active compound (**8d**) to evaluate its drug-likeness. Most of the predicted parameters were aligned with Lipinski’s rule. The drug-likeness scores and the bioavailability score of the tested compound (**8d**) were comparable to those of the reference drug (omeprazole), indicating that compound (**8d**) could be a potential drug-like candidate. However, the estimated aqueous solubility of the tested compound suggested that further optimization is needed by incorporating different polar and water-soluble groups into the core scaffold to improve the drug-likeness of these candidates.

## Data Availability

The original contributions presented in the study are included in the article/Appendix A, further inquiries can be directed to the corresponding authors.

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
