# Peer review of "Discovery of Benzopyrone-Based Candidates as Potential Antimicrobial and Photochemotherapeutic Agents through Inhibition of DNA Gyrase Enzyme B: Design, Synthesis, In Vitro and In Silico Evaluation"

_pharmaceuticals, 2024, doi:10.3390/ph17091197_

Round 1
Reviewer 1 Report
Comments and Suggestions for Authors
This article is fairly well written. I have some minor comments in order to improve the quality of the presentation.
- Figure 2: The image is too small, please expand it.
- Scheme 2: Usually the "Na metal" is just reported like "Na°" or just "Na".
- Table 1: I think the red color in the "Rs" is unnecessary.
- Molecular docking: Here I have a critical point: The compounds reported here could act as intercalating agents in the Bacterial DNA, like Ciprofloxacin. Why the author does not perform molecular docking using a PDB for the DNA-GyrProtein complex? For example: PDB: 2XCT, and reference: 10.1007/s11030-021-10302-7
Reviewer 2 Report
Comments and Suggestions for Authors
The manuscript is of general interest and might be suited for publication. However, there are many smaller issues which need to be addressed by the authors. It starts with the language quality, event standard term (e.g. genetic algorithm) are written incorrectly (“generic algorism” instead of “genetic algorithm”) suggesting that the authors do not have an in-depth knowledge of some techniques.
The abstract should be concise. Compound numbering should be avoided as it does not help at this stage.
Include one compound per line in Tables 1 and 3
Table 2 should be on one page to better allow comparison with the reference compounds. Moreover, the table is very wide, so that comparison of the values is not straightforward. “NA” should be replaced with “-“ or another small sign to better visualize the bacteria against which the compounds did not work. Information on standard deviation is missing.
Preferably, Table 2 should be changed to a heat map with different colors indicating different inhibition zones (Table with exact values could go to Supplementary). This would be much easier to grasp for the reader.
Atom naming such as C3 or C4 does not help if there is no figure showing this atom name.
l. 53/54. FDA-approved drugs (please remove FDA-, drugs are not only approved by this agency.
It is unusual not to write the reaction conditions in the captions and not into the schemes. Please change this.
IC50 values for enzyme inhibition are usually provided in molar, not in ug/mL. Please convert or include both values.
The authors provide NMR spectra, but no clear information on compound purity is included.
Information on the source of the reference compounds (xanthomyin, tobramycin, amphothericin B) is missing, as well as information on the purity.
Please do not write “1a,b” but rather “1a and 1b” and not 7a-c but 7a-7c
Section 2.4 should be renamed to “Binding mode investigation using molecular docking” or similar
It would be of interest to see the predicted binding mode of the reference ligand mode compared with the actual binding mode of the reference ligand.
The distance of the hydrogen bond (HB) in Table 4 is not clear. Is that the distance between the donor hydrogen and the acceptor atom? Please explain in the caption. Moreover, please round distances to values with one digit after the comma. The hydrophobic interaction is rather a van der Waals interaction, please correct.
Table 4: “Docking score” or “Binding energy” but not “Binding score”
In section 3.4.3 it was mentioned: “The docking protocol was validated through running initial docking experiments (pre-docking) for the reference ligand (II) and calculating the RMSD value.” However, there is no information on the result in the results part.
Moreover, in section 3.4.4 the sentence “…and the most stable conformational clusters were identified.” is unclear. What are “stable conformational clusters”? Did you just use the AutoDock clustering approach and selected the docking pose of the largest cluster? Please explain.
Molecular docking is not considered as “simulation”, please remove that word from the text.
Amino acid numbering should be “Val71” or “Val-71” and not “Val 71”.
There is no need to show both, Figure 4 and Table 3.
The labels of the residues in Figure 5 which are not shown (e.g. Val130) should be removed as this doe not help the reader. Labels should have the same style as in the text (VAL43 vs. Val 43). The carbon atoms of the ligands should be colored differently to better visualize them. At present, the color is almost identical to the ribbon of the protein. Labels should be placed in a way that allows easy reading, for example Ile78 is hard to read.
Section 2.5/3.5: IC50 values are determined based on a dose-response analysis using a broad concentration range, usually in a 1:2 or 1.3 dilution series. The author did not provide any information on the tested compound concentrations for determining the IC50. Moreover, dose-response curves are not provided. Please add them to the Supplementary information. In the methods part information in the dilution series and the starting concentration needs to be provided. In addition, the software used to determine the IC50 values needs to be mentioned and information on how the curves were fitted to the data points (similar to 3.2.4).
The references are not formatted correctly.
There are many other small issues which need to be corrected, including the language: poor, please review the manuscript carefully and ask for external help to improve the language. This is just a selection of the issues, it is by far not complete:
l. 31: Compounds 6a and b
l. 36/37: Compounds 8a-7
l. 48: The abrupt proliferation of microorganisms (this is a weird formulation)
l. 84: there seems to be several blanks after “and”
l. 91: same as l. 84 (after “The”)
l. 288: Lamarckian
l. 309: hydrogen bonding
l. 324: “H-binding interaction” does not exist
l. 346 showed
l. 345/346: “iconic” – there are no iconic interactions
l. 364: Compounds 8a, 8b, and 8d
l. 679: “Gasteiger charges were applied to merge the non-polar hydrogens using AutoDock Tools 1.5.6” – this is not correcvtly formulated, adding charges does not merge the hydrogens. Please rephrase.
l. 685: MolProbity (also citation needed)
l. 685/686: “AD4 parameters and Gasteiger charges were assigned to the protein atoms” – which AD4 parameters do you mean?
l. 689: genetic algorithm
l. 691: RMSD explanation missing
l. 710: E coli in italics
l. 729: “A series of” “was designed”
l. 739: …compounds to form strong hydrogen bonds with
l. 741/742: Three is something missing in “Compounds 8a-f should the required coplanarity”
l. 742: Docking scores do not indicate stability of binding, you mean the affinity or binding strength
Comments on the Quality of English LanguageSee above
Reviewer 3 Report
Comments and Suggestions for Authors
This manuscript describes the authors’ design, synthesis, and evaluation of compounds as potential antimicrobial and photo-chemotherapeutic agents.
A very important and somewhat obvious issue is the solubility of these compounds as they consist of mostly aromatic and hydrophobic groups. This could significantly affect the interpretation of assay results. Is the compound soluble at the concentrations tested? I’d like to urge the authors to provide evidence. Data presented in Table 3 and Table 5 should also have units of mol/L as opposed to g/L for easier comparison.
The authors’ claims that these compounds have a synergistic mechanism (as opposed to additive) and first-in-class are also problematic as I cannot see evidence of either synergy (meaning the effect is better than a simple combination of DNA gyrase B inhibition and intercalation into DNA) nor treating in a disease model.
There are other less critical issues. For example, the authors' design of compounds should be part of Results instead of Introduction and analyses of docking poses and ADME predictions could be less detailed, highlighting only key outliers such as the >5 logP value.
Comments on the Quality of English LanguageMinor errors including but not limited to in Line 51 ("or" should be "and") and Line 741 ("should" should be "showed").
Reviewer 4 Report
Comments and Suggestions for Authors
Elk-Haleem et al., present a characterization and biological testing of courmarin derivatives as putative DNA Gyrase inhibitors. Ifound the manuscript compelling for the most part. The authors present and convey information properly. There are, however, some comments and minor concerns:
Figure 5 is OK. However, I think that he diagram could benefit by being rendered with another visualization tool or by proving an alternative view of the binding site.
Similarly, Figure 6 conveys crucial information, yet the coloring scheme makes it difficult to follow, mostly by its contrast against a white background.
As for the NMR results in the supporting information, it would be useful to add the described structure, by these results.
Finally, while the results are indeed promising and experimental testing is robust. My suggestion would be to consider the addition of MMGBSA protocols of three compounds; ATP, 8d and 7d. In order to provide further data in the interactions made by flexible and rigid ligands.
Comments on the Quality of English Language
There are some editing and grammar errors in the manuscript. Please make a careful and through revision
Round 2
Reviewer 2 Report
Comments and Suggestions for Authors
There are still things to change, in particular the IC50 curves are missing.
Response to comment 11 by the authors: All the reference compounds (xanthomyin, tobramycin, amphothericin B) used were purchased with pharmaceutical grade > 98%.
My response: fine, but where is that information in the manuscript?
Response to comment 15 by the authors: The changes were applied to Table 4 and the other parts of manuscript including the captions. An informative caption was added to describe the hydrogen bonding (HB). Additionally, the hydrophobic interaction was replaced with van der Waals interaction (vdW). The HB distances were rounded to one decimal place.
My response: the description is not clear. What do you understand with hydrogen bond donor (the hydrogen atom or the connected heavy atom). I guess it’s the hydrogen atom of the donor site. Then please write it that way.
Response to comment 22 by the authors: Figure 5 was reconstructed to address the current comment and the comments from other reviewers. Labels were manually added to ensure clarity and to match the amino acid numbering format used throughout the manuscript. Unnecessary amino acids were removed to avoid confusion. The background color scheme was changed to make the compound atoms more visible. Labels were placed near the corresponding regions to facilitate easy reading.
My response: : The green and red labels are very hard to read. Also the number should be round to one digit after the comma.
Response to comment 23 by the authors: Thank you for your comment and valuable feedback. The concentrations (dilution series) of both tested compounds and standard drugs have been included in results and discussion part (section 2.5) and materials and methods part (section 3.5). Both the starting concentration and the software used to determine and calculate the IC50 values were included in section 3.5 (materials and methods part).
My response: The requested dose-response curves from which the IC50 values were derived are still lacking along with information on data analysis (fitting). These need to be provided, at least in the Supplementary part.
Additional comment:
Table 5: it does not make sense to report SD with 5 digits after the comma if the IC50 is provided with 2. Please shorten.
Comments on the Quality of English LanguageCheck for minor errors.
Reviewer 3 Report
Comments and Suggestions for Authors
The authors did not address the important concern about solubility. They claimed that "The compounds were dissolved in DMF and showed good solubility" without giving any evidence. More importantly, there is no investigation of aqueous solubility, a very relevant matter, at all. Hence the data are questionable and should not be published in this journal.